# Disseminated Coccidioidomycosis to the Spine—Case Series and Review of Literature

**DOI:** 10.3390/brainsci9070160

**Published:** 2019-07-07

**Authors:** Dinesh Ramanathan, Nikhil Sahasrabudhe, Esther Kim

**Affiliations:** Loma Linda University Medical Center, Loma Linda, CA 92354, USA

**Keywords:** Coccidioidomycosis, spinal infection, spinal pain

## Abstract

Coccidioidomycosis is a fungal infectious disease caused by the *Coccidioides* species endemic to Southwestern United States. Symptomatic patients typically present as community-acquired pneumonia. Uncommonly, in about 1% of infections, hematogenous extra pulmonary systemic dissemination involving skin, musculoskeletal system, and meninges occur. Disseminated spinal infection is treated with antifungal drugs and/or surgical treatment. A retrospective review of medical records at our institution was done between January 2009 to December 2018 and we present three cases of spinal coccidioidomycosis and review the current literature. Disseminated coccidioidomycosis can lead to spondylitis that can present as discitis or a localized spinal or paraspinal abscess. Spinal coccidioidomycosis is typically managed with antifungal treatments but can include surgical treatment in the setting poor response to medical therapy, intractable pain, presence of neurological deficits due to compression, or structural spinal instability.

## 1. Introduction

Coccidiomycosis is a fungal infectious disease caused by the *Coccidioides* species endemic to Southwestern United States. Also known as Valley fever, it is caused by *Coccidioides immitis* or *Coccidioides posadasii*. These organisms survive well in areas of low rainfall, few winter freezes, and alkaline soil in Arizona, New Mexico, West Texas, San Joaquin Valley of California, and parts of Mexico and South America. The primary method of infection is through inhalation of aerosolized arthrospores, although rarely, infection from a direct cutaneous inoculation is possible. The majority of infections are asymptomatic and self-resolving. Symptomatic patients typically present as community-acquired pneumonia with symptoms of fever, rash, and flu like symptoms. Uncommonly, in about 1% of infections, hematogenous extra pulmonary systemic dissemination involving skin, musculoskeletal system, and meninges occur. Involvement of the spine can range from discitis and paravertebral soft tissue infection to vertebral body erosion and neural compression [1,2,3]. Extrapulmonary disseminated coccidioidomycosis with involvement of the spine, either localized or multiple segments is treated with antifungal drugs and/or surgical treatment. In this article, we present cases of spinal coccidioidomycosis treated at our institution and review the current literature.

## 2. Materials and Methods

We conducted a retrospective review of medical records of all patients treated for spinal coccidioidomycosis at our institution. We queried the medical records of all in-patient admissions in our institution between January 2009 and December 2018 for the diagnosis of spinal coccidioidomycosis. We retrospectively reviewed patient demographics, clinical characteristics, presenting symptoms, radiological features, management, and outcomes of all patients treated for spinal coccidioidomycoses. Radiographic features of infection in the spine including the location of infection, characteristics of magnetic resonance imaging (MRI), number of vertebral segments involved, and involvement of neural elements and paraspinal structures were studied. An Institutional Review Board was done at our institution. 

## 3. Results

A total of 373 patients of coccidioidomycosis (pulmonary or systemic disseminated) were treated from January 2009 to December 2018 at our institution. There were three cases (male = 3, 0.80% of patients) of coccidioidomycosis infection of the spine during the same period. All three patients presented with pain symptoms in the spine, and two patients had neurological deficits. We describe the presentation, management and outcomes in these cases. 

### 3.1. Case Illustration 1

A 28-year-old man presented with a one-month history of lower back pain, worsening shortness of breath, and intermittent fevers over a period of the month. He was initially treated with azithromycin at an outside facility, which failed to resolve his symptoms. The back pain was progressively severe and radiated to both lower extremities, limiting his ambulation. He had no history of sick contacts, travel, or history of exposure to tuberculosis patients. Initial treatment included empiric antibiotics and screening tests for HIV, tuberculosis with PCR, legionella, and a endemic mycosis serology panel that included histoplasmosis, blastomycosis, and coccidioidomycosis. A CT scan of the chest revealed lucencies throughout mid-thoracic spine with adjacent prominence of paraspinal soft tissues suggestive of osteomyelitis and discitis. MRI of the T spine revealed abnormal marrow enhancement seen with varying degrees of paraspinal soft tissue enhancement, the most significant being at T6 where diffuse marrow enhancement and vertebral height loss was seen (Figure 1a,b). A biopsy of the lesion confirmed the coccidioidomycosis (Figure 1c). He was initiated on antifungal therapy—voriconazole and amphotericin B—followed by surgical debridement and stabilization. 

He underwent bilateral T7–9 laminectomies and foraminotomies for decompression of the spinal cord. The abscess was identified and noted to be fibrous and adherent to the duramater. Caseating tissue was noted epidurally extending more in to the left lateral recess. Debridement and washout of all extraneous infected tissue was performed. Four days later, the patient underwent a transnasal approach for incision and drainage of pre-clival and retropharyngeal abscess. At 1-year follow-up, he continued to have moderate to severe axial sharp pain in the neck and lower back. An MRI demonstrated persistent marrow changes in lumbar spine and pelvis, as noted previously.

### 3.2. Case Illustration 2

A 62-year-old man with past medical history of pulmonary fungal infection presented with a seven-month history of back pain. He was initially diagnosed with coccidioides pneumonia seven years ago and was placed on a fluconazole long-term treatment, which was later discontinued by another physician due to renal adverse effects. He presented with symptoms of severe lower back pain, which exacerbated when sitting or lying down. The symptoms of pain were associated with weakness in the lower extremities and poor balance while walking. He had chronic non-radiating back pain. MRI of the lumbar spine revealed L1–2 discitis and osteomyelitis with a paraspinal abscess (Figure 2a). Antibiotics therapy and fluconazole were started preliminarily. An image-guided interventional biopsy demonstrated coccidioidomycosis infection. Neurologically, having motor deficits and severe pain with imaging confirming compression of neural elements warranted surgical treatment. A lateral approach to the lumbar spine was undertaken to perform corpectomy of the L1 and L2 vertebral bodies along with discectomy and insertion of an expandable cage with a morselized bone graft. This construct was reinforced with a posterior instrumented fusion extending two segments superiorly and inferiorly (Figure 2b). The intraoperative specimen showed coccidiodes spherules within the bone specimen, consistent with dissemination of the infection to the spine (Figure 2c). 

### 3.3. Case Illustration 3

A 54-year-old man with a past medical history of hypertension, C3–4 osteomyelitis, prevertebral/epidural abscess, and left septic knee who presented initially to his primary care physician with left upper extremity and lower extremity weakness, left knee pain, and lab results remarkable for elevated white blood cell count. Infectious disease workup and joint aspiration of the left sternoclavicular joint infection with coccidioidomycosis were confirmed. He was treated with intravenous fluconazole and vancomycin. The MRI and CT scan for evaluating the weakness revealed osteomyelitis/discitis of C3–4 and a focal epidural abscess (Figure 3a,b). In addition, he was diagnosed with a left septic knee and underwent aspiration of joint effusion. A repeat MRI of the cervical spine later revealed improvement in the size and inflammation of the epidural abscess, and the patient was discharged on IV liposomal amphotericin B 50 mg per day. A few weeks later, the patient presented to the emergency room with fever, tachycardia, and tingling in the bilateral upper extremities. MRI of the cervical spine revealed retrolisthesis of C3 and C4, spinal canal stenosis, and cord compression due to extension of the epidural abscess into the level of the C5 vertebra. Furthermore, extensive discitis and osteomyelitis with collection in both the epidural and prevertebral regions were noted. The patient was clinically noted to have motor weakness in the left upper extremity and was transferred to our institution for a higher level of care and management. The abscess was surgically treated with incision and drainage of the prevertebral abscess, a C4 corpectomy, C3–5 fusion with placement of a cage, and an anterior plate from C3–5. Figure 3a shows the Coccidiodes spherules in the ventral epidural abscess. During the hospitalization, he also underwent several debridement procedures and arthrocentesis of his infected left knee.

## 4. Discussion

Coccidioidomycosis is a fungal infection that is asymptomatic in a majority of patients. Patients with symptomatic infection generally present with isolated pulmonary involvement manifesting as a flu-like syndrome—headache, night sweats, rash, arthralgia, and myalgia. Disseminated infection via bloodstream and lymphatics is less common and is noted frequently in immunocompromised patients. Extrapulmonary disease to the musculoskeletal system has a predilection for the axial bones and commonly involves the vertebral bodies.

Musculoskeletal involvement is a feature in about half of the disseminated forms with about 2–3% of this population being symptomatic. Vertebral and neurological involvement seems to be more common among Asian and African-American populations. Spinal coccidioidomycosis should be diagnosed early to prevent the local spread of infection and involvement of neural elements. Clinically, vertebral coccidioidomycosis commonly presents with symptoms of back pain or neck pain with associated radiculopathy, sensory disturbances, and motor weakness [4]. Both symptomatic and asymptomatic pulmonary coccidioidomycosis infections can lead to spinal coccidioidomycosis, although it is more frequent in patients with symptomatic pulmonary infection [5]. Therefore, patients with symptoms of back pain and associated neurological symptoms such as motor weakness or sensory symptoms, with travel or residence history in endemic areas should be evaluated for coccidioidomycosis [1,6,7]. *Coccidioides* species titer serological tests must be obtained early and is helpful in establishing the diagnosis. IgM elevation is noted in 1–3 weeks of infection, and IgG elevation occurs in 2–28 weeks, with titers greater than 1 in 128 is suggestive of bone or joint involvement [3,5]. Erythrocyte sedimentation rates and C-reactive protein, though not sensitive or specific to coccidioidomycosis, are frequently elevated.

MRI, though not diagnostic, can demonstrate erosive defects in vertebrae and endplates and vertebral body collapse or paraspinal extension in more advanced cases. An MRI is more sensitive than a CT scan or plain radiographic films at detecting early changes due to infection. Typically, both active and necrotic lesions display T1 lengthening with contrast enhancement. Active lesions are hyper-intense in T2-weighted sequences. MRI cannot differentiate the findings in coccidioidomycosis, vertebral metastasis, tuberculosis, and other infectious diseases. Biopsy of vertebral lesions, generally CT-guided, to detect *Coccidioides immitis*/*posadasii* spherules is required for definitive diagnosis even with known involvement of other organ systems to rule out other alternative causes for spinal involvement. Several stains, including hematoxylin and eosin and periodic acid-Schiff stains (PAS), calcofluor white fluorescent, and Gomori methenamine silver, can identify the pathogen microscopically [1,6].

### 4.1. Medical Therapy

Spinal coccidioidomycosis is managed with pharmacologic and surgical treatment. All symptomatic patients with spinal involvement are initiated on antifungal therapy for a period of 12 to 18 months [1,2,8,9,10]. Untreated patients can progress to sepsis and eventual death, and therefore all patients require medical therapy.

Amphotericin has been used conventionally to treat coccidioidomycosis. Antimycotic treatment with azoles is more commonly being used as first line therapy to avoid the adverse effects of amphotericin, especially in cases with localized spinal infections. Fluconazole is most commonly used azole antifungal treatment for coccidioidomycosis. Other azole treatments include voriconazole and itraconazole. Voriconazole has shown to be effective in treating disseminated infection in contrast to fluconazole or itraconazole [5]. More recently, posaconazole has demonstrated efficacy with superior osseous penetration [11,12]. Immunocompromised patients and patients with meningitis are treated lifelong with azoles [1,7,10]. Patients who respond poorly to azoles will them typically progress to polytherapy that includes amphotericin in addition to multiple azoles. Compliance with multiple medical therapy is a challenge in treating these patients, which is a major cause of relapse of infection, although progressive infection and decline are not uncommon while adhering to therapy [5,13,14,15,16,17,18,19,20]. 

### 4.2. Surgical Treatment

Surgical treatment is indicated in patients with severe disease or who are poor respondents to medical therapy or have intractable pain, the presence of neurological deficits, compression, or structural spinal instability [1,5,9,21,22,23]. Surgical treatment with medical therapy is noted to be effective in alleviating pain and symptoms and limiting the disease compared to medical therapy alone [2,5,9,24,25]. Titanium hardware has been safely used for spinal stabilization in coccidioidomycosis as with other spinal infections. These have been demonstrated to be resistant to biofilm formation and aid in eradication of infection by limiting spinal mobility in the involved segments [1,26,27,28,29,30,31]. We have summarized the treatment details and relevant outcomes in some of the larger case series (with greater than 10 patients) reported in the literature (Table 1).

Interval MRI should be used to assess for patients’ response to therapy. Surgical treatment is indicated in patients with spinal instability, neurological deficits or motor weakness, abscess formation, or osteolysis. Poor response to antifungal therapy or continued intractable pain with an attributable focus of infection are also additional indications for surgery. Surgical treatment involves debridement of infected tissue and instrumented fusion.

## 5. Conclusions

Disseminated coccidioidomycosis can lead to spondylitis that can present as discitis or localized spinal or paraspinal abscesses. Early diagnosis with serologic tests helps in preventing disease extension and systemic complications. Medical management with antifungal therapy is the first line of treatment for disseminated spinal coccidioidomycosis. In cases of spinal involvement leading to mechanical instability, compression of neural elements, and intractable pain, surgical treatment with debridement and/or instrumented stabilization is indicated. 

## Figures and Tables

**Figure 1 brainsci-09-00160-f001:**
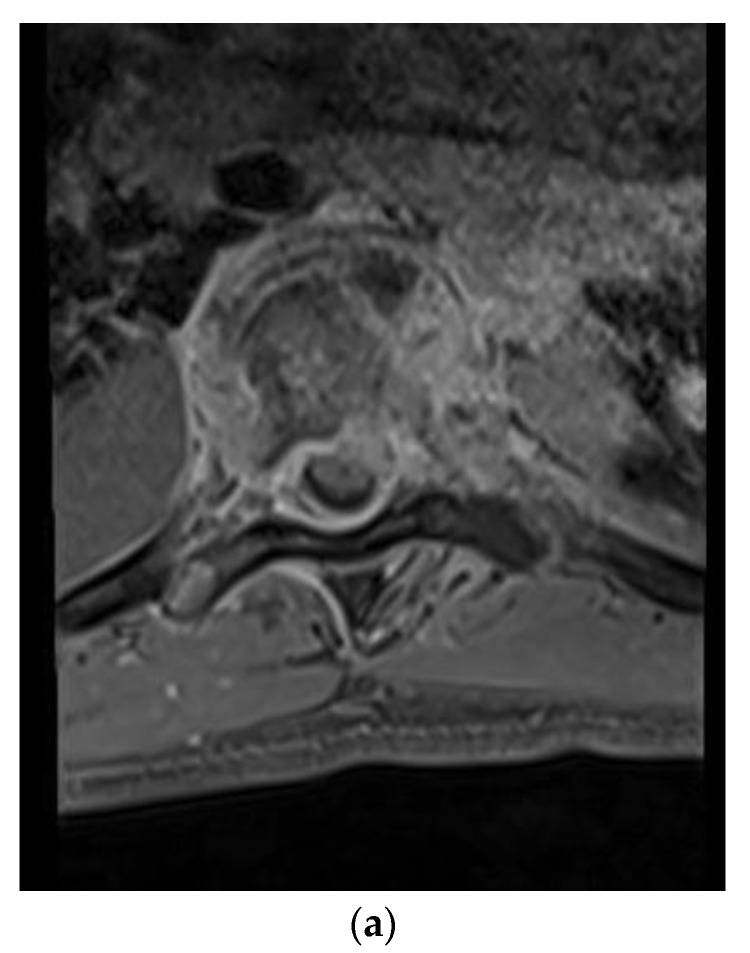
(**a**) Axial T1-weighted magnetic resonance (MR) images with contrast showing extensive vertebral body and soft tissue enhancement with compression of the spinal canal at T6. (**b**) Sagittal T1-weighted MR Images with contrast showing extensive enhancement throughout the vertebral bodies and soft tissue but most significantly at T6–7. **(c)** Hematoxylin and eosin (H&E) staining of the thoracic bone specimen showing acute osteomyelitis with abundant coccidioides organisms. Multinucleated giant cells with engulfed coccidioides spherules is a characteristic finding. Abundant acute inflammatory changes was noted in the marrow cavity.

**Figure 2 brainsci-09-00160-f002:**
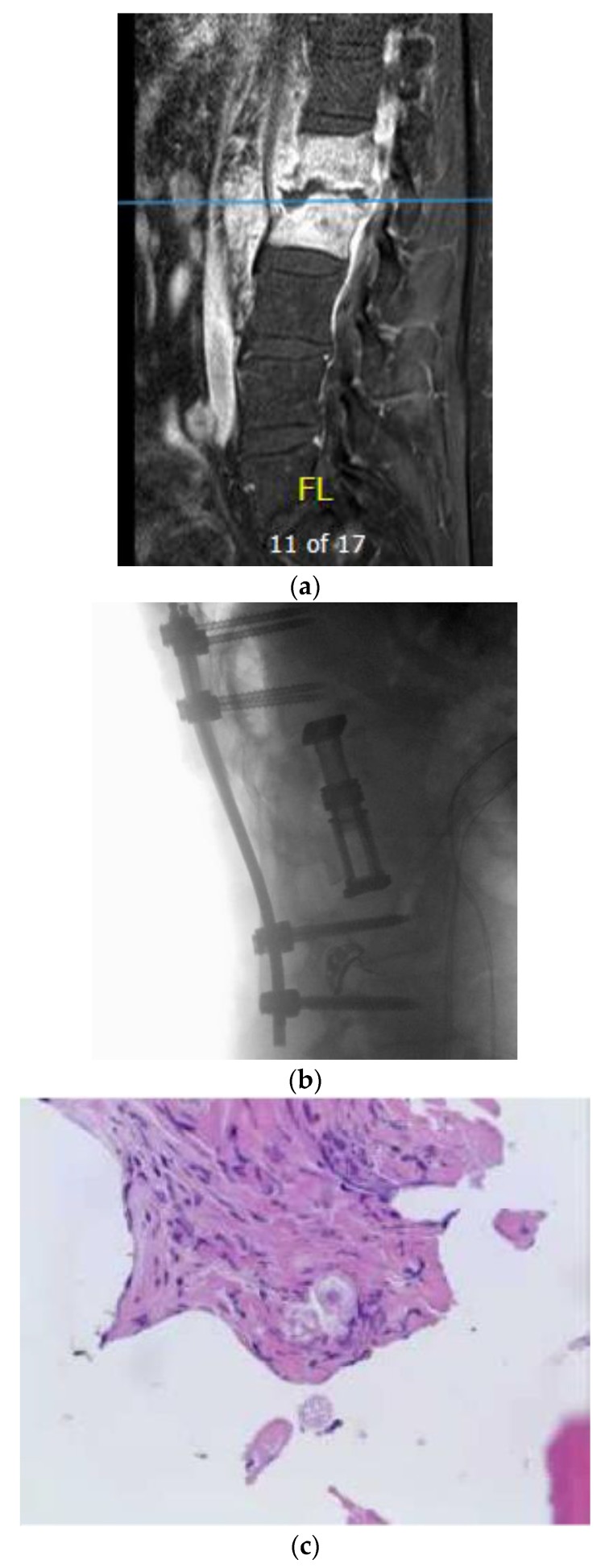
(**a**) Sagittal T1-weighted MR images with contrast showing extensive disc space destruction at L1–2 with epidural enhancement. (**b**) Sagittal T–L junction X-ray showing postoperative changes with a L1–2 corpectomy and instrumented fusion. (**c**) H&E staining of the intraoperative vertebral body specimen demonstrating the spherules within bone indicating osteomyelitis.

**Figure 3 brainsci-09-00160-f003:**
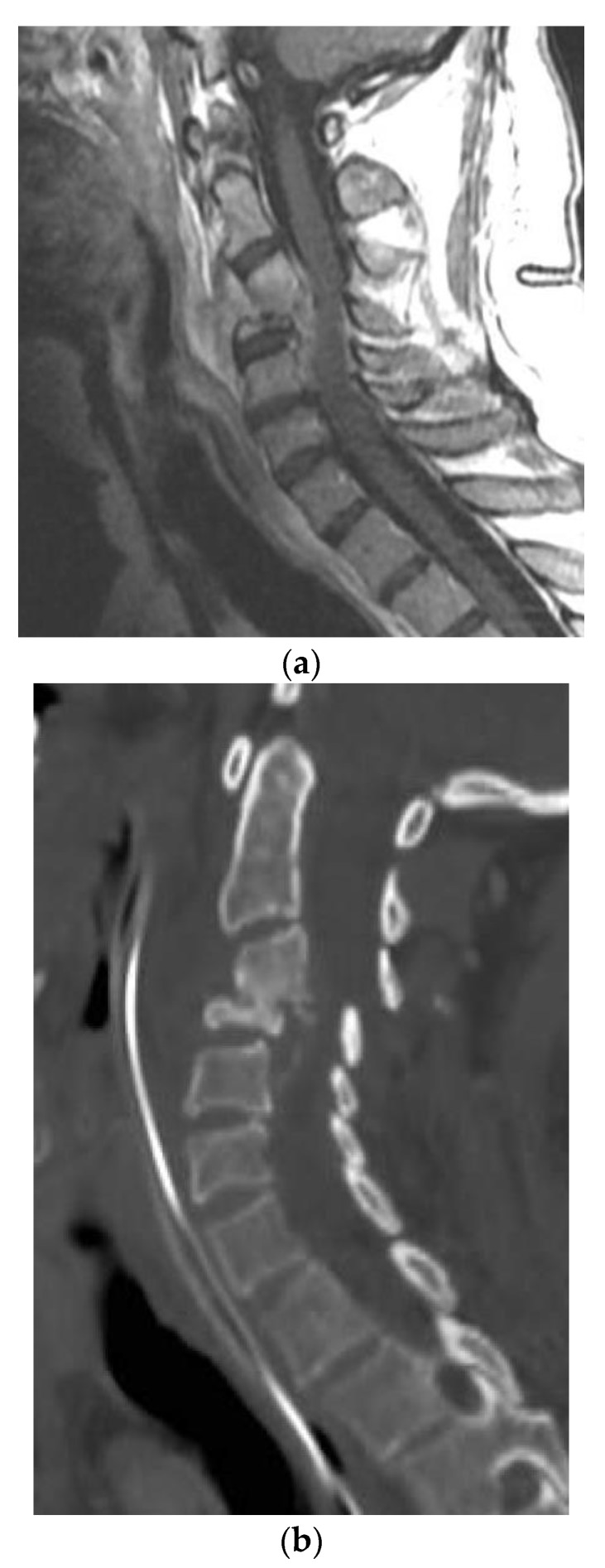
(**a**) Sagittal T1-weighted MR images with contrast showing extensive disc space destruction at C3-4 with epidural enhancement. (**b**) Sagittal cervical spine CT showing postoperative erosion of the C4 vertebral body significant retrolisthesis of C3 onto C4. (**c**) H&E stain of the cervical spinal pathologic specimen demonstrating the *Coccidioides* spherules within soft tissue, which is morphologically compatible with coccidioidomycotic osteomyelitis.

**Table 1 brainsci-09-00160-t001:** Summary of case reports of coccidioidomycotic osteomyelitis.

Study	No. of Patients	Medical Treatment Used	Surgical and Medical Treatment (No. of Patients)	Mean Age/Range (year)	Outcomes
Winter et al., (1978) [18]	12	IV amphotericin	Surgical + medical	2–35	Two patients died—one died five years later of coccidioidal meningitis, one died of fulminant spinal infection. One patient had paraplegia from thoracic spondylitis.
Zeppa et al., (1996) [20]	10	High dose IV liposomal amphotericin B. IV antibiotics with vancomycin and Zosyn for presumed bacterial infection	Surgical + medical (1)	33	Successfully treated then developed recurrence two years later despite being on suppressive oral antifungal treatment.
Herron et al., (1997) [25]	16	IV amphotericin B	Medical only group, medical + surgical group	40	Nine patients who had surgical and medical treatment had remission. Others were in medical only who had lost to follow-up.
Wrobel et al., (2001) [19]	23	Amphotericin and/or fluconazole	Surgical and medical	9–62	One: worsened postoperatively. One: reoperation needed. Four: died, 2/2 to fungemia. Most of the 15 surviving patients needed long-term antifungal treatment for extraspinal and spinal foci.
Kakarla et al., (2011) [9]	27	Amphotericin B, fluconazole, or voriconazole	Medical and surgical treatment	41.4	Follow-up for 19/27 patients: 16 improved 1 stable 1 worsened 1 died
Szeyko et al., (2012) [5]	39	All patients received triazole and 20 also received amphotericin B (usually early in the course)	Medical and surgical	35	None of the patients developed recurrence or refractory infection at the site of debridement. Six patients relapsed after stopping antifungal treatment.

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
