# Peer review of "Disseminated Coccidioidomycosis to the Spine—Case Series and Review of Literature"

_brainsci, 2019, doi:10.3390/brainsci9070160_

Reviewer 1 Report

The authors describe 3 cases of vertebro-spinal coccidioidomycosis in a unique center.

Please precise the city and country in the affiliation:  Loma Linda University Medical Center

Abstract 

line 16: what means “with pharmacologic treatment “

Line 27 change spores for arthrospores

Line 29: Please rephrase: “Majority of infections (up to sixty percent), they are asymptomatic and self-resolving”

Line 34: What means “Disseminated spinal infection”: spinal infection multifocal or not, after dissemination? Please clarify.

M&M 

It could have been nice to get records of the number of Cocci cases on the same period and to have the % of spinal involvement out of the total number of cases or hospitalized cases.

Results

Please replace endemic mycotic serology by endemic mycosis serology

In all cases: 

Please confirm how the biopsy confirmed the diagnosis of coccidioidomycosis. Pictures could be nice to add.

Please explain “PCP”

They authors should better discuss in the cases if spinal involvement is secondary to vertebral involvement or primary involvement. In the discussion also. This is not the same pathophysiology (dissemination to the CNS vs. locoregional invasion) in both cases. The title should also better reflect this.

Author Response

Please precise the city and country in the affiliation:  Loma Linda University Medical Center 

- Corrected

Abstract 

line 16: what means “with pharmacologic treatment “ - clarified antifungal treatment

Line 27 change spores for arthrospores - Done

Line 29: Please rephrase: “Majority of infections (up to sixty percent), they are asymptomatic and self-resolving” - Changed language

Line 34: What means “Disseminated spinal infection”: spinal infection multifocal or not, after dissemination? Please clarify. - Clarified that the this is Extrapulmonary disseminated infection to the spine

M&M 

It could have been nice to get records of the number of Cocci cases on the same period and to have the % of spinal involvement out of the total number of cases or hospitalized cases. - Done

Results

Please replace endemic mycotic serology by endemic mycosis serology - Done

In all cases: 

Please confirm how the biopsy confirmed the diagnosis of coccidioidomycosis. Pictures could be nice to add. - Done

Please explain “PCP”- Unclear which line this is front

They authors should better discuss in the cases if spinal involvement is secondary to vertebral involvement or primary involvement. In the discussion also. This is not the same pathophysiology (dissemination to the CNS vs. locoregional invasion) in both cases. The title should also better reflect this. - Changed the title to reflect this and noted in the discussion that this is due to disseminated infection that is extrapulmonary.

Reviewer 2 Report

Discussion section

Regarding the merits of MRI. You mention superiority of MRI over CT. CT actually is superior with regards to bony/vertebral involvement. MRI however is superior in detecting early disease, local soft tissue extension, and neurologic compression or involvement of the leptomeninges. MRI has over 92% sensitivity and specificity in detecting infection: acute pyogenic vs atypical. 

With regard to metastatic disease vs infection, the latter shows disc involvement on MRI images. Neoplastic metastasis usually does NOT involve the disc.

Author Response

I absolutely agree with the reviewer and changed the language to reflect that MRI is superior for early infection.